# A New Form of Taoist Theurgy in the Qing Dynasty: Xizhu Doufa in the Taoist–Tantric Fusion Style

**Yuhao Wu**

Department of Philosophy, Nanjing University, Nanjing 210023, China; william5yuhao@gmail.com

**Abstract:** The Longmen Xizhu Xinzong 龍門西竺心宗 was a Taoist sect that was active during the Qing Dynasty. The sect reportedly originated in India and has long been renowned for its Xizhu Doufa 西竺斗法. However, due to its secrecy and lack of literature, its true form remains a mystery. Examining the self-reported history of the Longmen Xizhu Xinzong, it can confidently be stated that Xizhu Doufa was often used in conjunction with Dharani and had deep roots in Tantric Buddhism. During the Sui and Tang Dynasties, Tantric Buddhism gained popularity in China and evolved into Tang Tantrism (Ch. Tangmi 唐密). There is a large amount of Dipper Method (Ch. Doufa 斗法)-related content in Tang Tantrism, which is a variant of China's original Dipper Method observed in India. After being passed back to China, it was named "Western Transmission". Many of the existing documents on the Dipper Method from the Ming and Qing Dynasties originated from Tang Tantrism. In terms of belief, they reflect the combination of Doumu 斗姆 and Marici; in terms of methods, they incorporate multiple elements, including the Taoist Thunder Method (Ch. Leifa 雷法) and Marici mantra. Overall, the ideas, lineage of transmission, and other aspects recorded in these documents are extremely similar to those emphasized by the Longmen Xizhu Xinzong, providing evidence for the origin of Xizhu Doufa. This evidence also provides a new reference for the direction in which Tang Tantrism developed after the Huichang Persecution of Buddhism 會昌毀佛.

**Keywords:** Longmen Xizhu Xinzong; Dipper Method; Tang Tantrism; Taoism in the Qing Dynasty

## 1. Introduction

The Longmen Xizhu Xinzong (the Heart Lineage of India) 龍門西竺心宗 was a Taoist sect that was active during the Qing Dynasty (1636–1912), famous for its Xizhu[1] Doufa[2] 西竺斗法, a secret theurgy from India. According to the records in *The Jingai Xindeng*[3] (金蓋心燈, abbr. *JGXD*), The sect originated in India and was introduced into China by Jizu Daozhe[4] 雞足道者 (fl. 13th–18th century).[5] Jizu Daozhe, who originally had no name, called himself Yedaposhe 野怛婆闍, which means "seeker of the Tao". He went to Beijing to visit Wang Changyue[6] 王常月 (?–1680), the high priest of the Longmen Sect[7] 龍門派 in the reign of Shunzhi 順治 (1643–1661). After being named Huang Shouzhong 黃守中 by Wang and converting to the Longmen Sect, Jizu Daozhe established his base at Jizu Mountain[8] 雞足山 and took on many disciples, thus spreading Xizhu Doufa. In the era of Qianlong 乾隆 (1735–1796), Min Yide 閔一得 (1749–1836) visited Jizu Daozhe with the *"Great Precept Book"* (Ch. *Da Jie Shu* 大戒書). Jizu Daozhe was delighted with the book and wanted to exchange it with Xizhu Doufa. Therefore, Min Yide stayed at Jizu Mountain for 3 months, and, after finishing his study, he compiled 12 volumes of *The Dafan Xiantian Fanyin Douzhou* (大梵先天梵音斗咒, abbr. *DFFYDZ*), which described Xizhu Doufa in detail (see Min et al. 2020, pp. 209–10).

Min Yide, the 11th-generation disciple of the Longmen Sect of Quanzhen Taoism, was a prominent figure in the history of Taoism in the Qing Dynasty. Due to his weak physical condition from childhood, Min Yide was ordered by his father to join the Longmen Sect at the age of 9 so as to learn methods to improve his health. He served as the main leader of the Taoist group on Jingai Mountain 金蓋山 in Huzhou 湖州 and achieved great success

in Taoist cultivation. However, he is also renowned for his systematic sorting of the Taoist doctrine and history. His collection of works, *The Gushu Yinlou Cangshu* (古書隱樓藏書, abbr. *GSYLCS*), comprises over 20 pieces that he either wrote or compiled, which mainly explain the principles and methods of Taoist cultivation. Additionally, the *JGXD*, which contains many invaluable resources and a large amount of information, has great significance for those studying Taoist history in the Qing Dynasty, as previously mentioned. Almost all the existing historical records of the Longmen Xizhu Xinzong were written by him.[9]

Min Yide's records suggest that the history of the Longmen Xizhu Xinzong can be traced back to Buddha's time (see Min et al. 2010, pp. 491–92). In his writings, Min Yide referenced *The Vasudhara Dharani* 持世陀羅尼經, a Buddhist sutra that is closely linked to the history of the Longmen Xizhu Xinzong. This sutra was translated into Chinese by Master Xuanzang 玄奘 (602–664) in the fifth year of the Yonghui 永徽 era (654). In addition to Xuanzang's version, there are two other Chinese translations of *The Vasudhara Dharani*, one by Master Amoghavajra (Ch. Bukong 不空; 705–774) and the other by Master Fatian 法天 (? –1001). All three versions are included in the *Taishō Tripiṭaka* 大正新脩大藏經 within the Tantric Buddhism category.[10]

*The Vasudhara Dharani* explains that a layman named Sucandra (Ch. Miao Yue 妙月) asked the Buddha for methods of supporting the poor, curing diseases, and fighting disasters. Then, the Buddha taught him Dharani (T20, n1162).

Min Yide said that, after learning Dharani, Sucandra taught it to many people and was regarded as the master of the second generation of the Jiatuo Zhengzong 伽陀正宗. The Jiatuo Zhengzong was the predecessor of the Longmen Xizhu Xinzong. For thousands of years, many successors, such as the Holy Monk Tieniu[11] 鐵牛聖僧, Bodhisattva Miaofan 妙梵菩薩, inherited the Jiatuo Zhengzong. In the Yuan Dynasty (1271–1368), Yedaposhe entered China as the master of the 100th generation and stayed at Jizu Mountain to await the appearance of a successor. Then, in the reign of Qianlong, Min Yide came to this place. (see Min et al. 2010, pp. 491–92).

The Longmen Xizhu Xinzong, with its long history, combines the essences of Taoism and Buddhism, showcasing a unique style. The sixth volume of the *JGXD* contains 13 biographies of individuals from the Longmen Xizhu Xinzong, spanning from Jizu Daozhe to Zhuzhu Sheng 住住生.[12] These 13 individuals are known for their formidable spiritual abilities but often exhibited unconventional behaviors that were quite different from those observed in the orthodox Longmen Sect of Taoism.

According to Min Yide's records, Jizu Daozhe was only proficient in Xizhu Doufa when he first arrived in China. Jizu Daozhe placed great importance on Xizhu Doufa, referring to it as "the treasure of India", and its requirements for successors were very strict (see Min et al. 2020, pp. 209–10). Although Xizhu Doufa was effective and highly exotic, its specific content is now difficult to determine, possibly due to its being heavily guarded. Additionally, an examination of Min Yide's book catalog reveals no work entitled *DFFYDZ*. What is the real face of Xizhu Doufa? It has become a mystery.

Due to the secret nature of the Longmen Xizhu Xinzong, there are few relevant studies on this sect. In the contemporary era, *An Example of Daoist and Tantric Interaction during the Qing Era: The Longmen Xizhu Xinzong* 清代道教と密教—— 龍門西竺心宗 by Monica Esposito from Italy placed the Longmen Xizhu Xinzong into the background of the interaction between Taoism and Buddhism in the Qing Dynasty and made it clear that the Longmen Xizhu Xinzong was the product of the integration of the Longmen Sect and Tantric Buddhism (Esposito 2005). *A Preliminary Study of the Longmen Xizhu Xinzong in the Qing Dynasty* 清代道教龍門西竺心宗初探—— 兼論雲南雞足山佛教對道教的影響 by Professor Sun Yiping 孫亦平 of Nanjing University studies the establishment, development, inheritance, religious beliefs, and cultivation characteristics of the Longmen Xizhu Xinzong from multiple perspectives, discusses the impact of Buddhism in Jizu Mountain on Taoism, and explains the new features of the Longmen Sect in the Qing Dynasty with the integration of Buddhism and Taoism, as well as the impact of the Longmen Xizhu Xinzong on Jiangnan

Taoism ([Sun 2015](#)). These studies encompass a wealth of materials but mainly focus on discussing the social background, historical inheritance, organizational form, and character deeds without delving into Xizhu Doufa directly. Therefore, while these studies inspired this paper to a great extent, new research directions still need to be explored. This article aims to first determine the basic attributes of Xizhu Doufa and then explore its form and connotations from the perspectives of Dipper belief and the rites associated with Xizhu Doufa, as well as its significance in Taoist theurgy during the Qing Dynasty.

## 2. The Basic Form of Xizhu Doufa

As *The Jueyun Benzhi Daotong Xinchuan* (龍門正宗覺雲本支道統薪傳, abbr. *JYDTXC*)[13] records:

> Master Min Yide obtained Xizhu Doufa and compiled *DFFYDZ*, passing it down to Master Fei Boyun 費撥雲. Fei shared it with other masters including Zhou Yifan 周抑凡, Ling Xiaohu 凌曉湖, and Chen Muzhai 陳牧齋. Then, the practice was inherited by Master Bian Dingsan 卞鼎三 and has remained consistent through the generations. Even until now, Xizhu Doufa still maintains its powerful effect of granting wishes. During the practice of the ritual, once the master prays, an immediate response will be received. The title of "India's Treasure" is truly well deserved.[14]

Here, it is revealed that Xizhu Doufa works through the form of "prayer". Additionally, it shows that the Xizhu Doufa lineage remained well organized after Min Yide. Throughout the biographical records of these masters, there are varying degrees of information regarding Xizhu Doufa. For instance, the *JYDTXC* recorded that Chen Muzhai utilized Xizhu Doufa to pray for clear skies and rain, cure illnesses, and ward off evil ([Hu et al. 1994](#), p. 472, vol. 31). Before he passed away, he even instructed his disciple Bian Dingsan to perform the Xizhu Doufa ritual:

> One day, Master Chen Muzhai was critically ill and instructed his disciple Bian Dingsan to quickly perform the Xizhu Doufa ritual, but it was not completed. Chen Muzhai said, "My retribution has come. Tonight, the crows and sparrows are chirping, just like the situation when my teacher Fei Boyun was critically ill and performed this ritual. My illness will definitely not be cured." Later, it turned out to be true.[15]

According to this record, Xizhu Doufa can pray for sunshine and rain, cure diseases and exorcism, and even fight disasters and prolong life. It is of the same nature as the Dipper Method (Ch. Doufa 斗法) that originally existed in China and belongs to the category of ritual in Taoism.

In Taoism, the Dipper Method developed from the belief in the Dipper. The Dipper belief is a major component of star worship in China and has a long history that dates back to primitive times. The earliest archaeological evidence related to the Dipper belief found in China is a rock painting discovered in Ji County, Shanxi 山西吉縣, that dates back approximately 10,000 years ago (see [Cultural Bureau of Linfen Administration in Shanxi Province 1989](#)).[16] Rituals corresponding to the Dipper belief also had early origins due to the latter's widespread popularity. Legend holds that the ancient king Yu 禹 created the primitive witchcraft known as Yubu 禹步, which imitated the arrangement of the Big Dipper's seven stars in a dance-like manner. *The Fifty-Two Prescriptions* 五十二病方 discovered in Mawangdui 馬王堆, believed to date from the pre-Qin era 先秦(Paleolithic era–221 BC), also documented a method of using Yubu to treat illnesses with a process highly similar to later Taoist practices (see [Wang and Wang 2014](#), p. 189). *The Xijing Zaji* 西京雜記 also recorded the ritual of praying to the North Star[17] for longevity in the imperial palace of Emperor Gaozu in the Han Dynasty 漢高祖 (256 BC/247 BC–195 BC), which proved that the ritual of praying to the stars was quite popular in the Qin (221 BC–207 BC) and Han (202 BC–220) Dynasties ([Ge and Zhou 2006](#), p. 146). In *The Record of the Three Kingdoms*

三國志, there is a record stating that Sun Quan 孫權 (182–252) asked Taoist priests to hold a star rite in order to extend the life of General Lü Meng 呂蒙 (178–220) (Chen and Pei 1982, pp. 1279–80). These examples demonstrate the longstanding history of the Dipper Method in China.

However, Xizhu Doufa, which is associated with the "Xizhu" crown, has its own distinct characteristics and is not entirely consistent with the original Dipper Method of China. Through a literature review, we can identify the unique inheritance of the Longmen Xizhu Xinzong.

As the predecessor of the Longmen Xizhu Xinzong, the name of Jiatuo Zhengzong itself reveals a great deal of information. In Chinese, the word "Jiatuo" mainly has two meanings. One is a translation of the Sanskrit word agada, which means good medicine. The other is a translation of the Sanskrit word gāthā, which means a rhythmic sentence.

As for the word "Zhengzong", it means orthodox school in Chinese. Jiatuo Zhengzong can be understood as the Zhengzong of Jiatuo in English. Due to the emphasis of the Longmen Xizhu Xinzong on Dharani, the Zhengzong of Jiatuo signifies the meaning of the orthodox school of gāthā or the orthodox school of Dharani (Ch. Tuoluoni 陀羅尼). The original meaning of Dharani is tantamount to remember and never forget. It is used as the general name for a kind of memory method, which is related to the process of cultural transmission in ancient India. The ancient Indians had language but no written characters, and the spread of knowledge mainly depended on word of mouth. In this context, Indians created many efficient methods of memory, and the Buddhist gāthā is one of these products. This highly rhythmic sentence is catchy and easy to remember. After the emergence of characters, this kind of memory method of the ancient Indians still survived, but its meaning and use have gradually changed. In Buddhism, the combination of Dharani and vidyā is a perfect example. This combination renders Dharani, which was originally a mere method of memory, increasingly mysterious.

Vidyā, which can be regarded as a kind of incantation, originated from the prayer used in primitive religion and is widely attested in Indian folk and Brahmanism. It is different from the meaning of Dharani for memory. At first, the vidyā was not allowed to be used in Buddhism. It was after the time of Tantric Buddhism that the combination of Dharani and vidyā became more common.

Furthermore, during the Yuan Dynasty, Buddhism in India and the Western Regions (Ch. Xiyu 西域) evolved into the form of Tantric Buddhism. If Jizu Daozhe did indeed come to China during the Yuan Dynasty, the teachings he brought with him would likely have belonged to this form of Buddhism. According to Min Yide, Jizu Daozhe's chanting was described as sounding like thunder and a bell, with a voice likened to metal and the tides of the sea (Min et al. 2010, p. 492), all of which are characteristics of Tantric Buddhism. In China, the Dipper Method is often accompanied by incantations, thus suggesting that the Dharani of Jizu Daozhe would align with the Xizhu Doufa practice. Additionally, Master Sucandra is also closely associated with Tantric Buddhism, considered as the first king of Shambhala in Kālacakra Tantra, and the Buddha taught the Kālacakra sutras at his request. From this perspective, it is highly probable that Xizhu Doufa was spread through Tantric Buddhism.

## 3. The Dipper Method in Tang Tantrism

At the turn of the Sui (581–618) and Tang (618–907) Dynasties, Buddhism entered a new stage of development, and Tantric Buddhism became very popular, forming Tang Tantrism (Ch. Tangmi 唐密) in China. There is a large amount of content on the Dipper Method in Tang Tantrism, which is obviously influenced by Taoism. According to Hsiao Teng-fu 蕭登福, there are more than 30 Buddhist sutras related to the worship of stars (Teng-fu Hsiao 1993, pp. 73–74). Most of these texts are the products of Tantric Buddhism of the Tang Dynasty, mainly based on the relevant ritual of the Dipper.

For example, in *The Foshuo Beidou-qixing Yanming Jing* (T21, n1307 佛說北斗七星延命經, abbr. *BDYMJ*), at the beginning of the sutra, there are images of the seven gods of the Dip-

per and corresponding secret signs, which are clearly derived from Taoism. The sutra then proceeds to match the seven gods of the Dipper with the Seven Buddhas. It is believed that, by receiving this sutra, one can gain many benefits, such as capacities for averting misfortunes, warding off ghosts, and attaining wealth. Hsiao Teng-fu believes that the concept of this sutra comes from *The Taishang Xuanling Beidou Benming Yansheng Zhenjing* (太上玄靈北斗本命延生真經, abbr. *BDJ*) and *The Gexiangong Li Beidou Fa* (葛仙公禮北斗法, abbr. *GXGLBDF*) (Teng-fu Hsiao 1993, p. 113).

The *GXGLBDF* is placed after *The Fantian Huoluo Jiuyao* (梵天火羅九曜, abbr. *FTHLJY*) in the form of an attachment. The *FTHLJY* (T21, n1311) is entitled "Compilation by Master Yixing[18] 一行 (683–727)". This text describes the good and bad luck of the time in which the Navagraha[19] 九曜 passed through each person's Mansion of Twenty-Eight Mansions[20] 二十八宿 and gives examples of the relevant rituals of the Navagraha, such as orientation, offerings, and mantra. This text shows the distinctive characteristics of Sino-Indian integration, in which the mantra and some images are Indian in style, but the main contents of the worship of the Dipper and related rituals have clear Chinese marks.

As for the *GXGLBDF* itself, Gexiangong, in its title, provides clues as to the origin of the text. Gexiangong is an honorific name referring to the legendary Taoist figure Ge Xuan 葛玄 (164–244). At the beginning of *The Beidi Qiyuan Ziting Yansheng Mijue* (北帝七元紫庭延生秘訣, abbr. *BDYSMJ*), a Taoist scripture from the Six Dynasties period (222–589), it is stated that "in the second year of Chiwu 赤烏 (239) in the Wu kingdom, Gexiangong received the scripture from the Lord Lao Zi; later, Mr. Ye passed it on to the world during the Wei Dynasty (220–266)"[21]. These two texts may have come from the same lineage. In terms of content, both Taoist scriptures follow the same principles of operation, with the *GXGLBDF* taking a more simplified approach.

The *GXGLBDF* is a short book, with only 600+ Chinese characters in total. This scripture vigorously promotes the importance of the Dipper to humanity and records a brief ritual in which offerings are made and prayers are directed towards the Dipper. In the *GXGLBDF*, only the mantra[22] at the end of the text probably has an Indian or Buddhist color, and all the rest reflect traditional Chinese knowledge of the worship of the Dipper.

The *GXGLBDF* seems unimpressive, but it has great influence and is valued by many scholars. For example, Hsieh Shu-Wei 謝世維 believes that the relatively definite Dipper ritual comes from the *GXGLBDF*, originating around the eighth century (Shu-Wei Hsieh 2018, p. 28). Hsiao Teng-fu believes that, by studying books such as the *FTHLJY*, *The Beidou Qixing Humo Fa* (北斗七星護摩法, abbr. *BDHMF*), *The Qiyao Xingchen Biexing Fa* (七曜星辰別行法, abbr. *QYBXF*) compiled by Master Yixing, and the sutras translated by Amoghavajra and Vajrabodhi (Ch. Jin'gangzhi 金剛智; 669–741), we find that the ritual and the star worship of Tantric Buddhism in the Tang Dynasty were deeply influenced by the *GXGLBDF* (Teng-fu Hsiao 1993, p. 104). In addition, Hsieh Shu-Wei also pointed out that the method of matching the hour of human birth to the seven stars of the Dipper in the *GXGLBDF* can also be found in *The Wuxing Dayi Yin Huangdi Dou Tu* 五行大義引黃帝斗圖, *BDYMJ*, *FTHLJY*, and *BDJ* (Shu-Wei Hsieh 2018, p. 30). Various studies have shown that there is an important source of the Dipper Method in Tang Tantrism, i.e., the Dipper Method passed down by Ge Xuan and recorded in the *GXGLBDF*. However, the Dipper Method passed down by Ge Xuan did not appear suddenly. It gathered many kinds of Dipper rituals that had already spread throughout China.

As for the *BDJ*, there are numerous disputes about the age of this scripture. Most of the ancients believed that this scripture was produced by Zhang Daoling 張道陵 (34–156), the founder of Zhengyi Taoism, while most contemporary scholars believe that it originated from the late Tang or early Song (960–1279) Dynasty. Although the various viewpoints are in opposition to each other, the materials supporting these viewpoints can mostly prove that the Dipper Method recorded in the *BDJ* is part of the same developmental context as the Dipper Method in the Taoist tradition and has the same origin as the Dipper Method in Tang Tantrism. For example, Hsieh Shu-Wei believes that the *BDJ* inherited the religious practice of the *GXGLBDF* (Shu-Wei Hsieh 2018, p. 32). Hsiao Teng-fu believes that the

*BDYSMJ*, which was probably handed down from Ge Xuan, was a scripture aiming to explain the ritual and offering of the sacrifice to the Dipper and should be viewed as an auxiliary text to the *BDJ*. Therefore, there is a clear inheritance chain of the Dipper Method passed down by Ge Xuan. The *BDJ*, *BDYSMJ*, and *GXGLBDF* are all links in the chain (Teng-fu Hsiao 1997, pp. 53–54). These views, regardless of whether the *BDJ* appeared early or late, all agree with on inheritance relationship with other scriptures of the Dipper. Hsieh Shu-Wei even believes that the rituals in the *BDJ* may be a part of folk religious practices which reveal some more primitive and more basic sources (Shu-Wei Hsieh 2018, p. 32). Meanwhile, the *GXGLBDF*, related to the *BDJ*, has directly affected the rituals, star worship, sacrifice, and other aspects of Tang Tantrism, which demonstrates the relationship between Taoism and Buddhism in the inheritance of the Dipper Method.

Many of the texts about star worship in Tang Tantrism were not translated but written or assembled by monks, being different from the general Buddhist sutras. For example, *The Qiyao Rangzai Jue* 七曜攘災訣 is entitled "Collected and Composed by the Brahmin Monk Jinjuzha from India"; the *QYBXF* is entitled "Written by Master Yixing"; the *BDHMF* is entitled "Written by Master Yixing"; the *FTHLJY* is entitled " Compilation by Master Yixing"; *The Xiuyao Yigui* 宿曜儀軌 is entitled "Written by Master Yixing"; *The Beidou-qixing Humo Miyao Yigui* 北斗七星護摩秘要儀軌 is entitled "Stated by the Acharya at the Translation Department of Xingshan Temple"; the *BDYMJ* is entitled "A Brahmin Monk Obtained in Tang China". This shows the strong correlation between these ancient books and Chinese culture, and many of these books directly inherited the concept and rituals of stars in China. Hsiao Teng-fu summarized six characteristics of the star worship literature in Tantric Buddhism, which generally apply to Tang Tantrism:

(1) In the Tantric Buddhist sutras, the gods of the Dipper and the Twenty-Eight Mansions gradually came to be interpreted as incarnations of the Buddha and Bodhisattvas.

(2) Tang Tantrism gradually came to mix Navagraha (excepting the two nodes of the Moon) with the Dipper as the same thing.

(3) In the ritual of worshiping stars, Tantric Buddhism is deeply influenced by Taoism.

(4) With regard to the motivation for worshiping the gods of stars, Tantric Buddhism shares the same goals as Taoism, such as prolonging life, bringing good fortune, praying for blessings, resolving difficulties, warding off evil spirits, and curing diseases.

(5) The legends of the gods of the Dipper and the Twenty-Eight Mansions descending to the human world gradually increased in number in Tantric Buddhism.

(6) Influenced by Taoism, Tantric Buddhism establishes a closer relationship between the gods of stars and human beings, and. in this tradition, the rituals used for dispelling disaster and praying for blessings became increasingly diverse (Teng-fu Hsiao 1993, the Conclusion of Chapter 2).

It should be emphasized again that the Dipper Method in Tang Tantrism was entirely based on the native Chinese belief and related rituals of the Dipper, without a direct affinity with Indian culture. Hsiao Teng-fu provided sufficient arguments for this statement, including two prominent reasons. Firstly, Buddhism regards gods as being still caught in the cycle of reincarnation and lower-level existences compared to Buddhas, Bodhisattvas, and Arhats; thus, it is impossible to worship stars such as the Dipper as gods. Secondly, ancient Indian astronomy was developed on the basis of the system of the Twelve Houses of the Zodiac, while ancient Chinese astronomy was developed on the basis of the Twenty-Eight Mansions as the main astronomical system. It is obvious that the Dipper Method in Tang Tantrism belongs to the latter system (Teng-fu Hsiao 1993).

After the process of being absorbed by Tang Tantrism and combined with the gradually improved Doumu (mother of the Dipper) 斗姆 Belief[23] after the Tang Dynasty, the Dipper Method developed in parallel and became interwoven with Buddhism and Taoism. From that time, the Dipper Method adopted very rich multicultural elements reflecting an obvious Taoist–Tantric fusion style.

## 4. The Gaodou Method and Longmen Xizhu Xinzong

During the Yuan (1271–1368), Ming (1368–1644), and Qing (1636–1912) Dynasties, a new type of Dipper Method emerged, which was known as the Gaodou (report to the Dipper) Method 告斗法. The Gaodou Method usually takes Doumu as the highest god for worship. By praying to Doumu's palace, it conveys information on topics such as the aversion of sickness, relief from misfortune, and transcendence of the dead. According to the research of Hsieh Shu-Wei and Tao Jin (Shu-Wei Hsieh 2020; Jin Tao 2012), there are several existing texts on the Gaodou Method, including the following:

- 先天雷晶隱書 *The Xiantian Leijing Yinshu*, abbr. *LJYS*, around 1353.
- 清微玄樞奏告儀 *The Qingwei Xuanshu Zougao Yi*, abbr. *QWZGY*, 1279–1368.
- 紫極玄樞奏告儀 *The Ziji Xuanshu Zougao Yi*, abbr. *ZJZGY*, 1279–1368.
- 清微灌斗五雷大法 *The Qingwei Guandou Wu-Lei Dafa*, abbr. *QWGDWLDF*, around 1382.
- 先天斗母奏告玄科 *The Xiantian Doumu Zougao Xuanke*, abbr. *DMZGXK*, 1445–1587.
- 梵音斗科 *The Fanyin Douke*, abbr *FYDK*, 1733.
- 先天大梵奏告金科 *The Xiantian Dafan Zougao Jinke*, abbr. *DFZGJK*, 1902.
- 亡斗節次 *The Wangdou Jieci*, abbr *WDJC*, 1924.
- 先天拔亡奏告科儀 *The Xiantian Bawang Zougao Keyi*, abbr. *BWZGKY*, 2000.

The names of these texts reflect a strong characteristic of inheritance, and some of them are very similar to the *DFFYDZ* of Xizhu Doufa. In addition, many clues related to the Longmen Xizhu Xinzong can be found in the lineage of the Masters recorded in some of these texts. Taking the *BWZGKY* as an example, the Masters mentioned in the text can be divided into six groups. Among them, the fourth group of "Masters of the Ziguang 紫光 (purple light) Sect" has the following masters:

- Master Jiaohua Ajiali 祖師紫光啟教西番教化阿迦利大法師.
- Saint Tieniu－Master Yixing 祖師鐵牛聖者一行禪師.
- Master Shizi Monk 祖師比丘三藏姚秦史紫真人.
- Master Lanniu Monk 祖師採石懶牛禪師.
- Master Holy Monk of the Southern Journey 祖師南渡聖僧法師.
- Master the Maiden who Conquers Demons 祖師降魔女仙師.
- Master Tiger head Monk 祖師虎頭三藏大法師.
- Master Puguang Monk 祖師比丘翰林學士普光真人.
- Master Bald Monk 祖師禿頭三藏大法師.
- Master Qingyuan 祖師清遠授宣教大法師.
- Master Yingguang Monk 祖師比丘應光大法師.
- Master Pishamen Bhikkhunī 祖師毘沙門比丘尼大法師.
- Master Huiguang 祖師慧光大法師.
- Master Yifeng 祖師一峰大法師.

The above Masters are basically Buddhist monks. According to Tao Jin's research, the Ziguang Sect was a branch of Tantric Buddhism in the Tang and Song Dynasties, which upheld the Marici mantra (Jin Tao 2014). Tao Jin has collected many folk documents, among which are records related to the history of the Ziguang Sect. For example, the preface to *The Dafan Chan* 大梵懺 records:

> In the later Qin Dynasty (384–417), there lived a monk named Jiaohua 教化[24] at Guta 古塔 (ancient pagoda) Temple. One day, a beggar girl came to the temple asking for alms. She was unkempt and gave off an unpleasant odor, causing everyone to despise her. However, Jiaohua was the only one who offered her help. Ten days later, the beggar girl said to him, "I am Marici. Today I give you the Brahma mantra. Practice hard and it will be effective." At the age of 180, Jiaohua passed the Brahma mantra on to Ajiali 阿迦利, Ajiali to Tieniu 鐵牛, Tieniu to Yixing, Yixing to Lanniu 懶牛,[25] and Lanniu to Bai Yuchan[26] 白玉蟾. As a result, Taoism inherited the Brahma mantra.[27]

Meanwhile, *The Xiantian Dafan Leishu* 先天大梵雷書, from the folk collection, records:

During his escape[28], Emperor Gaozong of the Song Dynasty 宋高宗 (1107–1187) saw a shining goddess in the sky who introduced herself as Marici. She told him that he was in trouble and that she was there to help. The emperor was overjoyed and expressed his gratitude to Marici by kowtowing to her. Later, he enshrined her portrait in the palace. One day, a monk who knew the Marici mantra appeared at the palace. After being recommended by the ministers, the emperor allowed him to stay. The monk was effective in praying for sunshine or rain and saving the suffering. One of the emperor's concubines fell seriously ill, and the emperor ordered the monk to treat her. As soon as he recited the mantra, the concubine regained consciousness. She said that while she was in a trance-like state, she had been bound by a demon king and unable to escape until seven flaming pigs appeared and burned the demon king, allowing her to escape. Finally, the emperor ordered the imperial concubine to inherit the monk's mantra and gave her the name "Maiden Who Conquers Demons (Ch. Xiangmo Nü, 降魔女)".[29]

On the basis of the legends documented in these texts, we can roughly trace the lineage of the Ziguang Sect, which is generally consistent with the fourth group's lineage in the *BWZGKY*. However, some inaccuracies have crept in with the passage of time. For instance, in the *BWZGKY*, Master Jiaohua and Master Ajiali are combined in a way that contradicts the legend in *The Dafan Chan*, although other texts present them as separate figures. Additionally, while other texts distinguish between Saint Tieniu and Master Yixing, the *BWZGKY* combines these two characters into one.

Saint Tieniu, also recorded in the *GSYLCS*, is called "Holy Monk Tieniu". He is the first person of the Longmen Xizhu Xinzong to be named after Sucandra. Saint Tieniu should have had an important position, but the historical data are missing. According to the above legend, Tieniu passed the Brahma mantra down to Yixing, who, as one of the founders of Tang Tantrism, was indeed associated with many legends of the Dipper and left many documents on the Dipper Method. This connection may indicate the source of Xizhu Doufa.

The key piece of information is that the Ziguang Sect and Longmen Xizhu Xinzong both believed that their sects were from the West, which, naturally, leads us to wonder whether they have the same source. However, by examining the above ritual texts, it can clearly be seen that this source is in China, which is completely in line with the development of the Dipper Method from the pre-Qin period.

According to Tao Jin's research, the earliest prototype of the Gaodou Method is the Doumu Method recorded in *The Daofa Huiyuan* (道法會元, abbr. *DFHY*) (Jin Tao 2013, pp. 501–5). The origin of the Doumu Method in the *DFHY* is based on the notion that Shangguan-Zhenren 上官眞人 combined the Thunder Method (Ch. Leifa 雷法) of the Shenxiao Sect[30] 神霄派 passed down by his uncle Wang Wenqing 王文卿 (1093–1153) with the Marici mantra passed down by Master Yixing. The existing Gaodou Method is mainly preserved within the Zhengyi tradition (Jin Tao 2012, p. 41).

Hsieh Shu-Wei also pointed out that the *LJYS* (around 1353) in the *DFHY* is the earliest text reflecting the combination of Doumu and Marici, and this combination was completed in the inheritance of religious rituals. This kind of religious ritual, which Hsieh Shu-Wei called the Doumu–Marici Method, comes from the tradition of the Shenxiao Sect, and its theory is inherited from the knowledge system of Wang Wenqing and Bai Yuchan (Shu-Wei Hsieh 2014, pp. 209–40).

From this information, we can see that the prevalent religious rituals centered on Doumu during the Ming and Qing Dynasties can be traced back to a common source: the ritual tradition of Doumu–Marici represented by the Doumu Method in the *DFHY*. This ritual tradition already existed and had spread among the people for a long time before the publication of the *DFHY* text. Therefore, it would be unjustified to classify this form

of the Dipper Method as simply Taoism or Tantric Buddhism. However, the underlying logic of the Dipper Method is inherently Chinese and has naturally evolved from China's longstanding tradition of star worship from the time of the pre-Qin period. This tradition may have even spread to India and then returned to China due to the popularity of Tantric Buddhism, with Master Yixing and Saint Tieniu possibly playing pivotal roles in this process. This forging of a brand-new style of ritual is characterized by a combination of the Thunder Method of the Shenxiao Sect with the mantras of Tantric Buddhism, which not only cultivates internal alchemy but also prays for sunshine and rain, cures diseases, exorcises evil, and manages disasters.

This reveals a significant fact, i.e., that, after the Huichang Persecution of Buddhism[31], Tang Tantrism did not disappear completely in China. Instead, it gradually shifted toward civilian development and maintained some communication with India. During the Song Dynasty, one branch of this sect merged into the Shenxiao Sect of Taoism and was known under the name of the Ziguang Sect. As another branch of inheritance, the Longmen Xizhu Xinzong, although the specific time of its infiltration is unclear, has been traced back to the Tang Dynasty through texts such as *The Vasudhara Dharani* and others. It was officially incorporated into Taoism during the Qing Dynasty. On the basis of these findings, we can establish the proposition that Tang Tantrism underwent a transformation into Taoism. However, further research is necessary to determine the specific composition and completeness of its form.

In addition to Xizhu Doufa, there was another kind of Dipper Method spread on Jingai Mountain. According to the *JGXD*, "When Chen Qiaoyun 陳樵雲 was 25 years old, he returned from Guangxi Province to Jingai Mountain and learned from Xu Longyan 徐隆岩 about the Ziguang Fandou[32] 紫光梵斗. He held rituals piously every day, and the sky dropped sweet dew for this"[33]. Xu Longyan, who teaches the Ziguang Fandou, was once the leader of the Taoist group of Jingai Mountain. The *JGXD* records that Xu Longyan was proficient in Taoist rituals. In his early years, Xu Longyan had converted to Zhengyi Taoism. Later, after arriving at Jingai Mountain, he passed on the Dipper Method to Chen Qiaoyun, Zhu Chunyang 朱春阳, Shi Changzai 史常哉, and others (Min et al. 2020, p. 157). This evidence shows that the Ziguang Fandou and the existing Gaodou Method all originate from the same source.

Furthermore, the *JGXD* also mentions that Chen Qiaoyun exchanged knowledge and skills with Li Chijiao 李赤脚, Zhang Pengtou 張蓬頭, Jin Huaihuai 金懷懷, and other members of the Longmen Xizhu Xinzong. This led him to become increasingly convinced that various Taoist sects, knowledge, and skills all originate from the same source (Min et al. 2020, p. 181). Min Yide and Chen Qiaoyun were both in charge of Jingai Mountain at different times and were on very friendly terms. Therefore, it is highly likely that Min possessed a deep understanding of the Ziguang Fandou. As a result, these records of Min Yide almost express the notion that the Ziguang Fandou shares the same origin as Xizhu Doufa.

The facts that the Ziguang Fandou spread through Zhengyi Taoism and that many members of the Taoist group of Jingai Mountain after Min Yide were effective in recording the Thunder Method of Zhengyi Taoism[34] reflect the communication between Quanzhen Taoism and Zhengyi Taoism in the Qing Dynasty.

## 5. The Historical Positioning of Xizhu Doufa

Through previous discussions, it is known that Xizhu Doufa was a new form of theurgy in Qing dynasty Taoism. This judgment has multiple layers of meaning.

Firstly, in an English context, Xizhu Doufa can be seen as a type of theurgy. The development of the Dipper Method progressed through a long process, with the early version mostly consisting of simple rituals for worshiping the Dipper. Later, the Gaodou Method integrated other practices such as internal alchemy cultivation, mantras, and rituals, rendering it far richer in content. According to the Gaodou Method, practitioners must reach a spiritual state of emptiness to summon various gods and goddesses, including the Nine Emperors of the Dipper and Doumu, even merging with them. This causes the Gaodou

Method to seem highly consistent with theurgy.[35] According to Min Yide's records, Xizhu Doufa contains very rich content. For example, after years of ascetic practices, Huosiren 活死人 was allowed to learn Xizhu Doufa. Jizu Daozhe first taught him Dana-yin 怛那印, which is a small part of Xizhu Doufa ([Min et al. 2020], pp. 219–20). Dana-yin is a kind of mudra (hand gesture) in Tantric Buddhism which can also be found in other Gaodou Methods (see Jin [Tao 2014]). The example of the teaching of Dana-yin illustrates the kinship between Xizhu Doufa and other Gaodou Methods, additionally highlighting the need for powerful personal qualities required for the practice of Xizhu Doufa (in terms of religious practice). In order to improve one's personal qualities, it is generally necessary to engage in Taoist internal alchemy cultivation or Tantric Buddhism yoga. There is ample information on the Tantra practice method of Marici in *The Great Marici Sutra* translated by Devaśāntika 天息災 (?–1000).

Secondly, Xizhu Doufa can be considered as a relatively "new" practice. From a chronological perspective, the Gaodou Method is the latest form of the Dipper Method to date and is still being passed down in contemporary times. Although Min Yide did not clearly classify Xizhu Doufa, the use of terms such as "Zougao" 奏告 and "Douke" 斗科 in the text indicate without a doubt that Xizhu Doufa belongs to the Gaodou Method.[36] Moreover, among the numerous Gaodou Methods, Xizhu Doufa is one of the newer ones. According to Hsieh Shu-Wei's research, early Gaodou Method texts either did not mention Doumu or did not fully integrate Doumu and Marici (Shu-Wei [Hsieh 2020]). However, the mentions of Doumu in the works of Min Yide and his disciples, as well as the thriving Doumu faith on Jingai Mountain, may indicate the mature Doumu beliefs contained within Xizhu Doufa.

From a formal perspective, Xizhu Doufa is very concise. This is not due to its primitivity but rather is a result of refinement and transformation. *The operate instruction of the Vasudhara Dharani* 行持佛說持世陀羅尼經法規則 written by Min Yide lists some precautions to be taken when conducting relevant rituals using *The Vasudhara Dharani*, and its emphasis on simplicity and directness is evident. For example, in the "Clothing" section, it states that "lay people can wear ordinary Taoist robes or dress neatly"[37]. *The Vasudhara Dharani* is the most important text in the Longmen Xizhu Xinzong aside from the *DFFYDZ*, and *The operate instruction of the Vasudhara Dharani* is said to have been handed down by Jizu Daozhe. Its stylistic tendencies should be consistent with those of Xizhu Doufa. From *The operate instruction of the Vasudhara Dharani*, it is evident that the doctrines and methods transmitted by the Longmen Xizhu Xinzong have been carefully edited and refined, abandoning cumbersome theory and taking a direct approach that is generally simple and straightforward. This is likely one of the reasons why the sect emphasizes the transmission of their teachings through the word "heart".

From a sectarian perspective, Xizhu Doufa is a unique blend of the Quanzhen Taoism and Zhengyi Taoism lineages. According to Hsieh Shu-Wei's research, the early Gaodou Method mainly derived from the Qingwei Sect 清微派 and Shenxiao Sect, combined with elements of the Golden Elixir Southern Sect 金丹南宗 and Tantric Buddhism (Shu-Wei [Hsieh 2020]). However, after the unification of various sects of Zhengyi Taoism in the Ming dynasty, the Qingwei Sect, Shenxiao Sect, and some of the Golden Elixir Southern Sect were incorporated into Zhengyi Taoism. Since then, the Gaodou Method has primarily been passed down within the Zhengyi Taoism sect, even in modern times. However, Xizhu Doufa belongs to the category of Quanzhen Taoism and blends multiple lineages. Reading Min Yide's works, elements of non-Quanzhen sects can be found throughout. This kind of fusion can be regarded as a breakthrough. After the passing of Min Yide, numerous branches developed from the Taoist group on Jingai Mountain. Many of these branches placed great emphasis on religious rituals, and most of their members were laypeople allowed to marry and have children. This style is remarkably similar to that of Zhengyi Taoism.



## 6. Conclusions

Xizhu Doufa's specific content is a mystery due to its secret transmission and lack of documentation. However, as a subset of the wider Dipper Method, Xizhu Doufa shares many traits with other types of Dipper Methods. Tracing the historical development of the Dipper Method can help us to understand Xizhu Doufa.

The Dipper Method originated from the belief in the Dipper. The worship of the Dipper has a long history in China, dating back to primitive society. It evolved into the belief in Doumu, which took at least a few thousand years. Eventually, Doumu merged with Marici, a process that took several hundred years, culminating in the fully developed Doumu–Marici belief. This belief spanned both Buddhism and Taoism and became a widely worshipped deity among the people, as well as the main object of veneration in the Dipper Method.

Before officially introducing Doumu as its main god, the Dipper Method underwent significant development. Yubu, which existed in primitive witchcraft, was an early form of the Dipper Method. After the birth of Taoism, the Dipper belief was strengthened and became more diverse. Among the various methods, the Dipper Method transmitted by Ge Xuan in the Wu region during the Three Kingdoms period (220–280) had a great influence, and many later schools of the Dipper Method traced their origins back to this template. During the Tang Dynasty, Tantric Buddhism became popular and formed Tang Tantrism in China. Many of the contents of Tang Tantrism were related to the Dipper Method, combining the original Chinese Dipper Method with Indian culture. As a result, after the Tang Dynasty, the Dipper Method mostly showed characteristics of the integration of Taoism and Tantric Buddhism.

Among the different methods, the Gaodou Method was a new type of Dipper Method established on the basis of the Doumu belief. In several existing Gaodou Method texts, the lineage of the Masters is quite remarkable. Taking the *BWZGKY* as an example, the Ziguang Sect recorded in the text has great similarities with the Longmen Xizhu Xinzong in terms of the Masters' lineage and practice methods. The Ziguang Sect was a Tantric Buddhism sect that practiced the Marici mantra during the Tang and Song Dynasties, and its origins can be traced back to Master Yixing. This sect was combined with Taoist sects such as the Shenxiao Sect and transmitted a Dipper Method that emphasized both internal alchemy cultivation and the cooperation of mantras, mudras, etc. This situation was closely related to the harsh, survival-threatening circumstances that Tang Tantrism encountered after the Huichang Persecution of Buddhism. After the Tang Dynasty, Tang Tantrism lost its complete organizational form in China. Nonetheless, through fragmentation, it either moved underground among the people or merged with Taoism. Following a lengthy period of integration, some parts of Tang Tantrism have even survived to the present day. From this perspective, the situation of the Longmen Xizhu Xinzong is basically similar to that of the Ziguang Sect. Therefore, Xizhu Doufa also blends elements of Tang Tantrism and the Taoist Thunder Method, and its basic form is still within the scope of the original Dipper Method practiced in China. When examining the transmission of Xizhu Doufa after Min Yide, mixtures with the Thunder Method of Zhengyi Taoism are often found, which serve as evidence of the Chinese descent of Xizhu Doufa.

**Funding:** This research received no external funding.

**Acknowledgments:** I would like to express my heartfelt gratitude to Yueqing Wang and Wenhua Shen for their strong support in writing this article.

**Conflicts of Interest:** The author declares no conflict of interest.

## Notes

[1]   The meaning of Xizhu is the country of India in the Western Region.

[2]   Doufa refers to the Dipper Method. "The Dipper" or "The Big Dipper" is also known as "The Plough" in English (Ch. Beidou, 北斗).

3    The *JGXD* was written by Min Yide and systematically records the history of the Longmen Sect of Quanzhen Taoism in the form of biographies. It is a comprehensive work on the history of the Longmen Sect before the mid-Qing Dynasty.

4    The meaning of Daozhe is a person who practices Taoism.

5    Min Yide recounted that when he encountered Jizu Daozhe, the latter was purportedly over 500 years old yet looked only as though in his 60s. Jizu Daozhe's eyes were bright and piercing while his voice resounded like a bell (see Min et al. 2010, pp. 575–76).

6    Wang Changyue (?–1680), a well-known Taoist in the late Ming and early Qing dynasties, was a seventh-generation master of the Longmen sect of Quanzhen Taoism. His greatest contribution was reviving the declining Taoism and revitalizing its spiritual essence; later, he was recognized as the founder of the revival of the Longmen sect of Quanzhen Taoism.

7    The Longmen Sect is a branch of Quanzhen Taoism established by Qiu Chuji 丘處機 (1148–1227). It not only inherits traditional Taoist thought but also reorganizes Taoist cultural treasures such as science, rituals, commandments, talismans, and elixirs. Its contributions have laid the foundation for Taoism today.

8    Jizu Mountain was originally the site of the venerable disciple Mahākāśyapa's practice. According to Buddhist scriptures, Mahākāśyapa was instructed not to enter Nirvana and to guard the Buddha's robe on Jizu Mountain until Maitreya Bodhisattva became a Buddha and passed the robe on to him. China's Jizu Mountain is located in the northwest of Binchuan County, Dali Bai Autonomous Prefecture, Yunnan Province. It is named after its three peaks at the front and one ridge at the back, resembling a chicken's foot. Jizu Mountain is now a holy site for the three branches of Buddhism: Theravada, Mahayana, and Tibetan Buddhism.

9    For more research on Min Yide, see Chen (2017).

10    Xuanzang's version: T20, n1162. Amoghavajra's version: T20, n1163. Fatian's version: T20, n1164.

11    The meaning of Tieniu is a bull that is as strong and tough as iron.

12    These 13 individuals are Jizu Daozhe, Guan Tianxian 管天仙, Dajiao Xian 大腳仙, Wang Xiuhu 王袖虎, Jin Huaihuai 金懷懷, Baima Li 白馬李, Zhang Pengtou 張蓬頭, Huosiren 活死人, Li Chijiao 李赤腳, Shizhao Shanren 石照山人, Li Pengtou 李蓬頭, Longmen Daoshi 龍門道士, and Zhuzhu Sheng 住住生.

13    After Min Yide's passing, his teachings were inherited and expanded upon by many disciples who established several branches centered around Jingai Mountain. One of these branches was the Jueyun Tan 覺雲壇, founded during the Guangxu period 光緒 (1875–1908). In 1927, Dai Benheng 戴本珩, Prime Minister of Jueyun Tan, led the compilation of the book *JYDTXC*. This book followed the style and content of the *JGXD* and provided additional details on the lineage and transmission of the sect following Min Yide's era.

14    自閔祖易得西竺斗法, 歸纂《大梵先天梵音斗法》, 傳之撥雲費師. 費傳周抑凡, 凌曉湖, 陳牧齋三師. 陳傳鼎三卞師. 師師相傳, 淵源一貫. 迄今斗法, 禱之輒應, 靈異卓著. 西竺至寶, 誠不誣也. (Hu et al. 1994, p. 430, vol. 31)

15    師一日病危, 命其及門卞子鼎三, 奏告急告斗科, 斗未竣, 師曰:"我報應已得, 鴉雀夜鳴, 與我師撥雲子病危告斗時同其報應. 我病必不起矣." 既而果然. (Hu et al. 1994, p. 472, vol. 31).

16    For further studies on the Dipper belief, see Zhu (2018).

17    Regarding the relationship between the Big Dipper and the North Star in ancient Chinese astronomy, please refer to Shen (2022).

18    Master Yixing (683–727), a monk of the Tang Dynasty, was known for his exceptional intelligence from a young age and had a comprehensive knowledge of various texts, particularly in the fields of calendrics, yin-yang theory, and the five elements. He learned from his teachers Subhakarasimha, Vajrabodhi, and Amoghavajra, inheriting and refining the teachings of Tantric Buddhism. Yixing made remarkable contributions to astronomy and calendrics and was called the "scientist among monks".

19    The Navagraha is a collective term for the nine celestial bodies in ancient Indian astrology, and the concept was introduced to China during the Tang Dynasty. The nine parts of the navagraha are the Sun, Moon, the planets Mercury, Venus, Mars, Jupiter and Saturn, and the two nodes of the Moon.

20    In ancient China, astronomers divided the ecliptic into four regions collectively known as the Four Symbols, and each was represented by a mystical animal. The Azure Dragon 青龍 symbolized the east, the Black Tortoise 玄武 represented the north, the White Tiger 白虎 stood for the west, and the Vermilion Bird 朱雀 represented the south. Each region contained seven mansions, adding up to a total of 28 mansions.

21    吳赤烏二年, 葛仙訪受之于太上老君. 至魏時, 葉先生傳之于世. (Jiyu Zhang 2004, p. 241, vol. 30)

22    唵萨缚诺刹怛罗(二合) 三磨曳室哩曳扇底迦俱噜娑婆(二合) 贺.

23    The development of the Doumu belief progressed through a lengthy process, with the fully matured belief integrating the Chinese Doumu and Indian Marici. This integration may have been completed as late as the mid-Yuan dynasty, but its groundwork may have been laid as early as the Tang dynasty by Master Yixing and others. For further research on the Doumu belief, please refer to the following sources: Hsiao (2004, 2011).

24    The meaning of " 教化" is to influence others through the means of education.

25    The meaning of " 懶牛" in Chinese is lazy bull.

26  Bai Yuchan (1134–1229), a Taoist priest of the Southern Song Dynasty (1127–1279), was an expert in internal alchemy theory and one of the five ancestors of the Golden Elixir Southern Sect 金丹南宗 in Taoism. He was also very skilled in poetry, calligraphy, painting, and other arts. It is generally believed that Bai Yuchan organized the religious group of the Golden Elixir Southern Sect and was the actual founder of this sect.

27  姚秦時, 古塔寺有西番僧名教化, 遇一貧女乞食於寺, 襤褸臭穢, 眾皆惡之, 惟教化憫而濟焉. 越旬日, 女曰: 我摩利支天, 梵音符咒授汝, 精煉無求不應. 教化壽至一百八十傳阿伽利, 利傳鐵牛, 牛傳一行, 行傳懶牛, 牛傳玉蟾先生, 玄門得其法焉. (Tao 2014)

28  Emperor Gaozong fled to the south to escape the Jurchen people and created the Southern Song Dynasty (1127–1279). This happened between 1127 and 1130 and involved him using political and military strategies to establish a new dynasty in the south with Lin'an 臨安 as the capital city. It's sometimes called the "difficulties of Lin'an".

29  宋高宗南渡之時 . . . 但見空中光茫燭天 . . . 曰吾乃大梵先天之女雷祖摩利支也, 帝今有難特來救護 . . . 高宗大悅, 再拜啟謝, 密記聖像繪彩于宮中奉祀. 一日, 忽有僧能持此咒, 大臣舉之, 敕留演奉. 凡祈禱雨陽, 拯救患難, 靡不感應. 忽有宮妃為祟所憑, 命僧治之. 念動梵音, 其妃即醒. 乃曰: 適被大魔所縛, 不能得脫, 恍惚之間見有七豬, 火焰迸身, 燒烙魔體, 遂得更生 . . . 帝敕原病宮嗣傳其法, 賜名降魔女. (Tao 2014)

30  The Shenxiao Sect is a branch of Taoism that originated in the late Northern Song dynasty (960–1127) and was prevalent during the Southern Song through to the Yuan and Ming periods. The lineage recognizes Chen Tuan 陳摶 (871–989) and Bai Yuchan as two of its founders, with Bai Yuchan having authored several works on the Thunder Method. Additionally, the Shenxiao Sect is considered as a sub-branch of the school of talismanic magic.

31  The Huichang Persecution of Buddhism refers to a series of policies initiated by Tang Emperor Wuzong 唐武宗 (814–846) during his reign from 840 to 846. The peak of this persecution was the edict issued in April of the fifth year of the Huichang era (845). However, after Wuzong's death in the sixth year of the Huichang era, the throne was inherited by the Tang Emperor Xuanzong 唐宣宗 (810–859), who reinstated the worship of Buddha, thus ending the persecution. This event dealt a severe blow to Buddhism in China.

32  The meaning of Fandou is Indian-style Dipper Method.

33  (陳樵雲) 二十五, 歸自粵西, 禮隆巖師於雲巢, 近求玄秘. 隆巖師授以紫光梵斗, 遂休雲巢. 日夜虔禮, 甘露為之屢降. (Min et al. 2020, p. 180)

34  For example, Ling Xiaohu has a record of being profficient in medical skills and the Thunder Method of Zhengyi Taoism in *The Daotong Yuanliu* (道统源流, abbr. *DTYL*), which also states that Bian Dingsan is profficient in all the kinds of Thunder Methods (Lay Buddhist Zhuangyan 1929).

35  Theurgy is a type of magic. It consists of a set of magical practices performed to evoke beneficent spirits in order to see them or know them or influence them, for instance, by forcing them to animate a statue, to inhabit a human being (such as a medium), or to disclose mysteries (Riffard 1983).

36  For information on the relationship between concepts such as "Zougao" and "Douke" with the Gaodou Method, please refer to (Shu-Wei Hsieh 2020).

37  凡庶只用道袍或衣冠俱可. (Min et al. 2020, p. 490)

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
