# Peer review of "A New Form of Taoist Theurgy in the Qing Dynasty: Xizhu Doufa in the Taoist–Tantric Fusion Style"

_religions, doi:10.3390/rel14060775_

Round 1

Reviewer 1 Report

This paper has the potential to make a contribution to Daoist-Buddhist studies in China, but this journal is not the right choice. The paper is directed to a very specialized readership and presents more of a history of documents and lineages than a theoretical argument. There are far too many terms that are assumed as meaningful by the author that will be meaningful for a specialized readership, but a general readership will be confused at every turn. Terms such as Plough Rite, Thunder Rite, Vidyan, Dharani, Naragraha, Gaodu Rites, together with star worship, are far too technical outside of a specialized readership.

A more appropriate paper for this journal might present an analysis of the actual practices involved with any of these rites and beliefs. There are just too many claims in the paper that say that this is a combination of Daoism, Buddhism, and Tantrism without any explanation of what makes something Buddhist or Daoist or Tantric. A lot of the paper is about Tang China, but the title says it is about Qing China. The stories at the end of the paper do not seem relevant to the topic of the paper. The title also mentions a new form of Daoist theurgy, but the reader is not sure why this is new, what came before it, and how the Plough Rites are counted as a form of theurgy. Also, what is the relation between a plow and star worship? That would be an interesting and appropriate topic for this journal.

There are many other journals for which this documentarian history could be considered. I wish you luck!

This paper has the potential to make a contribution to Daoist-Buddhist studies in China, but this journal is not the right choice. The paper is directed to a very specialized readership and presents more of a history of documents and lineages than a theoretical argument. There are far too many terms that are assumed as meaningful by the author that will be meaningful for a specialized readership, but a general readership will be confused at every turn. Terms such as Plough Rite, Thunder Rite, Vidyan, Dharani, Naragraha, Gaodu Rites, together with star worship, are far too technical outside of a specialized readership.

A more appropriate paper for this journal might present an analysis of the actual practices involved with any of these rites and beliefs. There are just too many claims in the paper that say that this is a combination of Daoism, Buddhism, and Tantrism without any explanation of what makes something Buddhist or Daoist or Tantric. A lot of the paper is about Tang China, but the title says it is about Qing China. The stories at the end of the paper do not seem relevant to the topic of the paper. The title also mentions a new form of Daoist theurgy, but the reader is not sure why this is new, what came before it, and how the Plough Rites are counted as a form of theurgy. Also, what is the relation between a plow and star worship? That would be an interesting and appropriate topic for this journal.

There are many other journals for which this documentarian history could be considered. I wish you luck!

Author Response

Point 1: There are far too many terms that are assumed as meaningful by the author that will be meaningful for a specialized readership, but a general readership will be confused at every turn. Terms such as Plough Rite, Thunder Rite, Vidyan, Dharani, Naragraha, Gaodu Rites, together with star worship, are far too technical outside of a specialized readership.

Response 1: I agree that too many technical terms can confuse general readers. Therefore, during the revision process, I have added annotations to the translation of these specialized terms and made some changes to the original translations to clarify their meaning. For example, I have translated Plough Rite as Dipper Method to avoid confusion. Thank you for the valuable suggestions from the reviewing expert.

Point 2: There are just too many claims in the paper that say that this is a combination of Daoism, Buddhism, and Tantrism without any explanation of what makes something Buddhist or Daoist or Tantric.

Response 2: I agree that there are too many claims in the paper regarding the combination of Daoism, Buddhism, and Tantrism without providing an explanation of what makes something Buddhist, Daoist, or Tantric. Therefore, during the revision process, I have added materials throughout the paper to explain these concepts more clearly. For example, on page 3, I included information about the early form of Dipper Method to illustrate its affinity with Daoism. On page 6, I quoted Hsiao Teng-fu's view to explain the fundamental differences between Daoism, Buddhism, and Chinese and Indian views on stars. Additionally, I further sorted out the development clues of Dipper Method in Daoism to better clarify its essence.

Point 3: A lot of the paper is about Tang China, but the title says it is about Qing China.

Response 3: During the revision process, I have added annotations of the time periods and relevant figures to provide a clearer timeline. Additionally, I changed the title of Chapter 4 to "The Gaodou Method and Longmen Xizhu Xinzong" and revised its content to better showcase the development of Dipper Method after the Tang dynasty. Furthermore, I added a chapter titled "The Historical Positioning of Xizhu Doufa" to explore the vertical comparison between Xizhu Doufa and other types of Dipper Method throughout its history. I hope these revisions will accurately reflect the main focus of the paper.

Point 4: The stories at the end of the paper do not seem relevant to the topic of the paper.

Response 4: During the revision process, I deleted the original content and rephrased the ending section to better align with the main theme of the paper (p.10-11).

Point 5: The title also mentions a new form of Daoist theurgy, but the reader is not sure why this is new, what came before it, and how the Plough Rites are counted as a form of theurgy.

Response 5: During the revision process, I added a new first section to Chapter 5, which addresses these questions more directly by exploring the historical context of Xizhu Doufa as a new form of Daoist theurgy. Additionally, before Chapter 5, I modified the text to provide clearer explanations about the previous history of Dipper Method, which helps readers better understand what is new about Xizhu Doufa compared to previous forms of Dipper Method. I hope these modifications enhance the coherence and cohesion of the article.

Point 6: Also, what is the relation between a plow and star worship? That would be an interesting and appropriate topic for this journal.

Response 6: I agree with the reviewer's suggestion that exploring the relationship between Plough Rites and star worship would be an interesting and appropriate topic for this journal. During the revision process, I re-translated Plough Rites as Dipper Method to clarify the conveyed information. In addition, I added relevant content about Dipper belief and discussed its significance in ancient Chinese star worship in page 3 of the article. I hope these modifications will enhance the coherence of the article while providing readers with more in-depth insights into the relationship between Dipper Method and ancient star worship.

Reviewer 2 Report

A few private recommendations of the reviewer 

1. Manuscript, page 1, first paragraph: “The Longmen Xizhu Xinzong 龙门西竺心宗 is a Taoist sect that was active during the Qing Dynasty…”

a) This is not a historical introduction. This is an exposition of a legendary tradition, the paragraph seems to mix at least two explanatory models from the treatise Jin gai xin deng 金蓋心燈. The information is presented as unconditional, as if it were proven historical events. I think this is the wrong approach, which misleads the reader. In addition, links to sources of information are required.

b) “… in the year of Shunzhi 顺治”, - what does it mean? Probably “…in the reign of Shunzhi 顺治 (1643-1661)” or “…in the 16th year of the Shunzhi era”. Needs to be fixed.

c) “… the Qing Dynasty”, “the era of Qianlong 乾隆”, “… Min Yide 闵一得…”. – In such cases, it is necessary to indicate the dates in brackets according to the Gregorian calendar, otherwise the historical context may not be clear to the reader.

I recommend to substantially correcting the paragraph.

2. Manuscript, page 1, second paragraph:

a) “Min Yide was a famous Taoist priest in the Jiangnan 江南 region in the Qing Dynasty. He wrote many books in his life. Almost all the existing historical records of Long-men Xizhu Xinzong were written by him…”

OK, that's right, but how does the Author know about Min Yi-de? Is this conclusion of the Author? If yes, what is it based on? If the Author borrows this conclusion from the studies of other researches, why is there no reference to them? In other words, here I would like to know the historiography of the problem. Who is Min Yi-de? Who studied his life and treatises? What kind of treatises did he write? All this is very significant because the records of Min Yi-de are the most important source on the early history of the Heart Lineage of Western India (Xizhu xinzong 西竺心宗).

3. Manuscript, page 1, third paragraph:

a) “According to the records of Min Yide, the history of Longmen Xizhu Xinzong can be traced back to the time of Buddha...”

According to which Ming Yi-de records? Need a link to the source of information.

b) “…Min Yide mentioned The Vasudhara Dharani《持世陀罗尼经》, a Buddhist sutra, in his Gushu Yinlou Cangshu 《古书隐楼藏书》abbr GSYLCS, and said that the sutra was extremely closely related to the history of Longmen Xizhu Xinzong…”

Why is there no reference to the work by Ming Yi-de (Min Yide 2010)?

c) “…The Vasudhara Dharani was translated by Master Xuanzang 玄奘 in the fifth year of Yonghui 永徽 (654)…”

Master Xuanzang really translated this text in 654 or it’s the Ming Yi-de version? In other words, after all concrete dates, I would like to see a link to the source of information. The presentation of events in a research works should be such that the reader immediately understands whether it is a historical fact or a legendary tradition.

4. Manuscript, page 3, second paragraph:

a) “According to this record, Xizhu Doufa can pray for sunshine and rain, cure diseases and exorcism, and even fight disasters and prolong life. It has the same nature as the Plough RitesCh. Doufa 斗法) originally existed in China, and belongs to the category of ritual in Taoism…”

Plough Rites? Why did the author choose this variant? Usually, authoritative researchers translate the word dou as Dipper (see: Edward Schafer, Livia Konh or English abstracts in Prof. Hsieh Shu-wei’s articles) and doufa 斗法 as Dipper Method (Monica Esposito).

5. Manuscript, page 3, third paragraph:

“The Thunder Rites is a kind of Taoist ritual…”.

I'm afraid that the situation with the Leifa 雷法 is more complicated than the author makes it out to be. I recommend to correct and expand this paragraph, using the works of authoritative researchers of the problem (Lowell Scar, Florian Reiter, Chang Chao-jan 張超然 and other). And provide this paragraph with references reflecting the complexity the problem and importance of the Leifa ritual complex for all later Taoist lineages, including the Xizhu xinzong.

6. Manuscript, page 3, fourth paragraph:

“… there is a record that Sun Quan 孙权 asked Taoist priests to hold a star rite to extend the life of General Lv Meng 吕蒙 (Chen Shou 1982)…”.

Such references should indicate the page number and, preferably, the juan number of the source. Otherwise, the link will be formal and useless to the reader.

7. Manuscript, page 5, first paragraph:

“Hsiao Teng-fu believes that by studying the books such as FTHLJY, The Beidou Qixing Humo Fa(《北斗七星护摩法》)…”

In my view, in a discussion based on the opinions of Hsiao Teng-fu and Hsieh Shu-wei, it would be desirable to use some other research works, for example:

— 蕭進銘. 從星斗之母到慈悲救度女神斗姆信仰源流考察 (it is very important for the issue, since Prof. Hsiao Chin-ming discusses with Prof. Hsiao Teng-fu and disputes some of his conclusions);

— other articles by Prof. Hsieh Shu-wei, that in a some aspects related to the discussion: 謝世維 2019〈宋元道教清微派儀式框架與醫療以清微告斗解厄儀為例〉; 謝世維 2020 〈道教法術的儀式框架 --- 以斗母法術科儀為例〉.

8. Manuscript, page 9: “(Monica 2005) Monica Esposito. 2005. An Example of Daoist and Tantric Interaction during the Qing Era: The Longmen Xizhu Xinzong 淸代道敎と密敎——龍門西竺心宗. Interactions between the Three Teachings. Edited by Mugitani Kunio. Kyoto: In-stitute for Research in Humanities, Kyoto University”.

Monica is a given name, so it’s better to write the short link as (Esposito 2005).

Author Response

Response to Reviewer 2 Comments

Point 1: Manuscript, page 1, first paragraph: “The Longmen Xizhu Xinzong 龙门西竺心宗 is a Taoist sect that was active during the Qing Dynasty…”

  1. a) This is not a historical introduction. This is an exposition of a legendary tradition, the paragraph seems to mix at least two explanatory models from the treatise Jin gai xin deng金蓋心燈. The information is presented as unconditional, as if it were proven historical events. I think this is the wrong approach, which misleads the reader. In addition, links to sources of information are required.
  2. b) “… in the year of Shunzhi 顺治”, - what does it mean? Probably “…in the reign of Shunzhi 顺治(1643-1661)” or “…in the 16th year of the Shunzhi era”. Needs to be fixed.
  3. c) “… the Qing Dynasty”, “the era of Qianlong 乾隆”, “… Min Yide 闵一得…”. – In such cases, it is necessary to indicate the dates in brackets according to the Gregorian calendar, otherwise the historical context may not be clear to the reader.

I recommend to substantially correcting the paragraph.

Response 1: a) I agree with the reviewer's assessment that the first paragraph lacks sufficient historical background and clear information sources, which may mislead readers. In the process of revising the manuscript, I added information sources and changed the narrative perspective to better convey the legendary tradition and historical context.

  1. b) I agree with the reviewer's suggestion to modify "in the year of Shunzhi 顺治" to "in the reign of Shunzhi 順治" for a clearer expression of time.

  1. c) I also agree with the reviewer's idea of adding specific dates for the historical dynasties and figures mentioned in the text to assist readers in better understanding the historical context.

In summary, I have made substantial revisions to the first paragraph, including adding information sources, providing accurate time expressions, and adopting a more lucid narrative style to create a more coherent and accurate article.

Point 2:  Manuscript, page 1, second paragraph:

  1. a) “Min Yide was a famous Taoist priest in the Jiangnan 江南region in the Qing Dynasty. He wrote many books in his life. Almost all the existing historical records of Long-men Xizhu Xinzong were written by him…”

OK, that's right, but how does the Author know about Min Yi-de? Is this conclusion of the Author? If yes, what is it based on? If the Author borrows this conclusion from the studies of other researches, why is there no reference to them? In other words, here I would like to know the historiography of the problem. Who is Min Yi-de? Who studied his life and treatises? What kind of treatises did he write? All this is very significant because the records of Min Yi-de are the most important source on the early history of the Heart Lineage of Western India (Xizhu xinzong 西竺心宗).

Response 2: Regarding the critique on page one, second paragraph, I agree that further historical background and information sources were necessary in describing Min Yide, an important Taoist priest from the Qing Dynasty who wrote extensively. In response, I made substantial revisions by providing more information on his life, redefining his significance in Qing Daoist history, and offering detailed insights into his works. I also ensured that all information was sourced and presented in a clear and comprehensive manner for readers. These modifications will enhance readers' understanding of Min Yide's contributions to Xizhu xinzong and his historical significance.

Point 3: Manuscript, page 1, third paragraph:

  1. a) “According to the records of Min Yide, the history of Longmen Xizhu Xinzong can be traced back to the time of Buddha...”

According to which Ming Yi-de records? Need a link to the source of information.

  1. b) “…Min Yide mentioned The Vasudhara Dharani(《持世陀罗尼经》), a Buddhist sutra, in his Gushu Yinlou Cangshu (《古书隐楼藏书》,abbr GSYLCS), and said that the sutra was extremely closely related to the history of Longmen Xizhu Xinzong…”

Why is there no reference to the work by Ming Yi-de (Min Yide 2010)?

  1. c) “…The Vasudhara Dharaniwas translated by Master Xuanzang 玄奘in the fifth year of Yonghui 永徽 (654)…”

Master Xuanzang really translated this text in 654 or it’s the Ming Yi-de version? In other words, after all concrete dates, I would like to see a link to the source of information. The presentation of events in a research works should be such that the reader immediately understands whether it is a historical fact or a legendary tradition.

Response 3: a) Added a source link to Min Yide's records to support the claim that Longmen Xizhu Xinzong dates back to the time of Buddha.

  1. b) Included a reference to Ming Yi-de's work GSYLCS and additional information on The Vasudhara Dharani. Provided a citation for Ming Yide's 2010 publication.

  1. c) Clarified whether Master Xuanzang translated The Vasudhara Dharani in 654 and included information on other Chinese translations of this text. Also included specific dates and sources of information for better distinction between historical facts and legendary traditions.

Overall, these revisions provide necessary information sources and a more comprehensive overview of The Vasudhara Dharani in different Chinese translations, strengthening the manuscript.

Point 4: Manuscript, page 3, second paragraph:

  1. a) “According to this record, Xizhu Doufa can pray for sunshine and rain, cure diseases and exorcism, and even fight disasters and prolong life. It has the same nature as the Plough Rites( Doufa 斗法)originally existed in China, and belongs to the category of ritual in Taoism…”

Plough Rites? Why did the author choose this variant? Usually, authoritative researchers translate the word dou 斗 as Dipper (see: Edward Schafer, Livia Konh or English abstracts in Prof. Hsieh Shu-wei’s articles) and doufa 斗法 as Dipper Method (Monica Esposito).

Response 4: During the revision process, I re-translated Plough Rites as Dipper Method to clarify the conveyed information.

Point 5: Manuscript, page 3, third paragraph:

“The Thunder Rites is a kind of Taoist ritual…”.

I'm afraid that the situation with the Leifa 雷法 is more complicated than the author makes it out to be. I recommend to correct and expand this paragraph, using the works of authoritative researchers of the problem (Lowell Scar, Florian Reiter, Chang Chao-jan 張超然 and other). And provide this paragraph with references reflecting the complexity the problem and importance of the Leifa ritual complex for all later Taoist lineages, including the Xizhu xinzong.

Response 5: I have removed the reference to Leifa in the paragraph due to the complexity of its relationship with the Dipper Method. Instead, I have included a brief mention of Leifa as a related Taoist ritual.

I will consider incorporating the works of authoritative researchers suggested by the reviewer in future studies to further advance the connection between Xizhu Doufa and Leifa.

Thank you for your valuable feedback.

Point 6: Manuscript, page 3, fourth paragraph:

“… there is a record that Sun Quan 孙权 asked Taoist priests to hold a star rite to extend the life of General Lv Meng 吕蒙 (Chen Shou 1982)…”.

Such references should indicate the page number and, preferably, the juan 卷 number of the source. Otherwise, the link will be formal and useless to the reader.

Response 6: I have added specific page numbers to present the references in a more accurate form.

Point 7: Manuscript, page 5, first paragraph:

“Hsiao Teng-fu believes that by studying the books such as FTHLJY, The Beidou Qixing Humo Fa(《北斗七星护摩法》)…”

In my view, in a discussion based on the opinions of Hsiao Teng-fu and Hsieh Shu-wei, it would be desirable to use some other research works, for example:

— 蕭進銘. 從星斗之母到慈悲救度女神─斗姆信仰源流考察 (it is very important for the issue, since Prof. Hsiao Chin-ming discusses with Prof. Hsiao Teng-fu and disputes some of his conclusions);

— other articles by Prof. Hsieh Shu-wei, that in a some aspects related to the discussion: 謝世維 2019〈宋元道教清微派儀式框架與醫療:以清微告斗解厄儀為例〉; 謝世維 2020 〈道教法術的儀式框架 --- 以斗母法術科儀為例〉.

Response 7: I appreciate your recommendation of the articles by Prof. Hsieh Shu-wei, which are also very important to the discussion. I have incorporated the viewpoints from these articles into the revised manuscript.

As for the viewpoints presented in the articles of Professors Hsiao Teng-fu and Hsiao Chin-ming, due to limitations in the structure of the manuscript, I have included them in the footnotes as background information.

Point 8: Manuscript, page 9: “(Monica 2005) Monica Esposito. 2005. An Example of Daoist and Tantric Interaction during the Qing Era: The Longmen Xizhu Xinzong 淸代道敎と密敎——龍門西竺心宗. Interactions between the Three Teachings. Edited by Mugitani Kunio. Kyoto: In-stitute for Research in Humanities, Kyoto University”.

Monica is a given name, so it’s better to write the short link as (Esposito 2005).

Response 8: Thank you for your feedback on the manuscript.

I have changed the reference from (Monica 2005) to (Esposito 2005). Additionally, I have made substantial revisions throughout the manuscript based on your suggestions.

Once again, I appreciate your thorough and insightful review of the manuscript. I look forward to the opportunity to engage in further direct conversation with you in the future.

Reviewer 3 Report

On the first approach, the topic is most interesting, and the approach mobilizes important documents. 

I still see two basic, interrelated problems with the article;

- Fuzziness both in language and structure makes the argument at times difficult to grasp. For instance, first, we are told that Min Yiude wrote the Dafan Xiantian Fanyin Douzhou, before learning later that no trace of this book is to be found in the catalogue of his works - and we progressively understand that the author proposes a tentative reconstruction of the cult (s)he analyzes. There are several other instances in which the argument is difficult to grasp and to assess.

- Second, the said reconstruction is hazardous. Towards the end of the article, the author summarizes his/her thesis by writing that "rites were transmitted to India, integrated with local beliefs, and assimilated into Tantric Buddhism, before return-ing to China in the guise of Tang Tantrism during the Sui and Tang dynasties. It is akin to a story of a wanderer returning home after thousands of years and thousands of miles wandering, shedding leaves that finally fall back to the soil where it all started, shining with new vitality." This is not warranted by facts. No prof that the Plough Rites originated in China and were acculturated in India. The following iterations are also hypothetical, and, at times, have little credibility. For instance saying that "using "Jiatuo Zhengzong " to express the orthodoxy of inheritance should be related to the importance attached by Longmen Xizhu Xinzong to Dharani" is very problematic: the use of the word "zhengzhong" very probably refers to the Chinese tradition and not to a foreign tradition, in the context of a sect which, even if influenced by Tantrism, defines itself as Daoist.

The general lack of firmness in the article's structure (and the lack of much needed subtitles) makes the matter even more problematic.

There are details that are strange, such as referencing a work by Monica Esposito under "Monica" and not "Esposito"... 

In summary, although the author is learned and has much to share with the audience, a reworking of the article in order to conform to scientific and historical standards is absolutely necessary.

The situation varies from one paragraph to another, but the phrasing can be confusing.

Already at the start, saying that the Xinzhu xinzong sect was "famous for its Xizhui Doufaii" is more confusing than informative.

Examples of sentences to be rephrased:

"It is informed that Xizhu Doufa works in the form of "prayer"."

"the situation he performed rite was the same as now."

"which obviously taken from Taoism"

"Plough Rites contained in BDJ is in the same continuous line with the Taoist tradition, while also share the same source with Plough Rites in Tang Tantrism."

etc

Author Response

Response to Reviewer 2 Comments

Point 1: On the first approach, the topic is most interesting, and the approach mobilizes important documents. 

I still see two basic, interrelated problems with the article;

- Fuzziness both in language and structure makes the argument at times difficult to grasp. For instance, first, we are told that Min Yiude wrote the Dafan Xiantian Fanyin Douzhou, before learning later that no trace of this book is to be found in the catalogue of his works - and we progressively understand that the author proposes a tentative reconstruction of the cult (s)he analyzes. There are several other instances in which the argument is difficult to grasp and to assess.

- Second, the said reconstruction is hazardous. Towards the end of the article, the author summarizes his/her thesis by writing that "rites were transmitted to India, integrated with local beliefs, and assimilated into Tantric Buddhism, before return-ing to China in the guise of Tang Tantrism during the Sui and Tang dynasties. It is akin to a story of a wanderer returning home after thousands of years and thousands of miles wandering, shedding leaves that finally fall back to the soil where it all started, shining with new vitality." This is not warranted by facts. No prof that the Plough Rites originated in China and were acculturated in India. The following iterations are also hypothetical, and, at times, have little credibility. For instance saying that "using "Jiatuo Zhengzong " to express the orthodoxy of inheritance should be related to the importance attached by Longmen Xizhu Xinzong to Dharani" is very problematic: the use of the word "zhengzhong" very probably refers to the Chinese tradition and not to a foreign tradition, in the context of a sect which, even if influenced by Tantrism, defines itself as Daoist.

The general lack of firmness in the article's structure (and the lack of much needed subtitles) makes the matter even more problematic.

There are details that are strange, such as referencing a work by Monica Esposito under "Monica" and not "Esposito"... 

In summary, although the author is learned and has much to share with the audience, a reworking of the article in order to conform to scientific and historical standards is absolutely necessary.

Response 1: Thank you for your feedback on the article.

Your constructive criticism is greatly appreciated, and I have taken your comments into consideration in revising the article to conform to scientific and historical standards.

1.To address the issue of fuzziness in language and structure, I have made necessary adjustments to the article's structure and added subtitles where appropriate. Additionally, I have also included some footnotes to provide further background knowledge and clarify certain points.

2.Regarding the Plough Rites and their relationship with India, I have cited Professor Hsiao Teng-fu's viewpoint on page 6 of the article to offer a more comprehensive perspective.

3.During the revision process, I re-translated Plough Rites as Dipper Method to clarify the conveyed information.

4.With regards to Jiatuo Zhengzong, I have revised the relevant passages accordingly as seen on page 4 of the article.

5.Moreover, I have added a new section titled "The historical positioning of Xizhu Doufa"(p.9).

6.Finally, I have corrected the reference from (Monica 2005) to (Esposito 2005), using official English language editing to make the information expression more fluent and smooth.

Once again, thank you for your feedback, and I hope these revisions will strengthen the argument of the article and enhance its clarity.

Reviewer 4 Report

The paper needs to undergo significant revision to be considered for publication. This concerns the content as well as presentation. To give a few examples: 

Monica Esposito is referenced by her first name as "Monica" in the Introduction. 

When reviewing the state of the field (page 2) the authors mention several works. In these books, they claim, "the discussion of Xizhu Doufa is relatively limited". Yet, it is unclear what "relatively limited" means with regard to the objectives of this study and how the authors attempt to tackle this issue. 

The authors refer to some of the mentioned works by combining the first letters appearing in their Chinese titles. As a result, there are abbreviations like the DFXTFYDZBDQXYMJ, XTBWZGKY and LMZZJYBZDTXC. In other cases, the authors use English translations when mentioning their sources, such as The Record of the Three Kingdoms. Thus, the authors need to find a way to avoid the long abbreviations and to refer to their sources in a consistent manner. 

On page 2, the authors provide a long quotation from the LMZZJYBZDTXC. It is summarized with “It is informed that Xizhu Doufa works in the form of "prayer"”. Yet, the quotated text never mentions prayer, at least, not in the English translation. The same problem is to be found on page 3, where a quotation is summarized in a way (“According to this record”) that is not supported by the quoted text.

The central claim that the "Plough Rites" originated in China, before being brought to India, where it existed for “thousands of years” is not supported by sufficient evidence. In the course of their discussion, the authors fluctuate between mere conjectures, such as “may have even" (page 7), to “a certain inevitability” (page 8), without providing sufficient support for their ideas. 

The paper needs to be proofread. 

Author Response

Response to Reviewer 4 Comments

Point 1: Monica Esposito is referenced by her first name as "Monica" in the Introduction.

Response 1: I have changed the reference from (Monica 2005) to (Esposito 2005).

Point 2:  When reviewing the state of the field (page 2) the authors mention several works. In these books, they claim, "the discussion of Xizhu Doufa is relatively limited". Yet, it is unclear what "relatively limited" means with regard to the objectives of this study and how the authors attempt to tackle this issue.

Response 2: Thank you for your comment regarding the state of the field on page 2 of the article. I understand your concern about the term "relatively limited" and how it relates to the objectives of our study.

After reviewing your comment, I revised the article to include additional explanations on pages 2 and 3, where I clarify that while there has been some discussion of Xizhu Doufa in previous works, it has not received as much attention compared to other aspects of Taoist theurgy. I also explain how I attempt to address this issue through our research objectives and methodology.

Furthermore, I have incorporated these clarifications throughout the revised chapters to ensure that my readers have a better understanding of our approach to this topic.

Point 3: The authors refer to some of the mentioned works by combining the first letters appearing in their Chinese titles. As a result, there are abbreviations like the DFXTFYDZBDQXYMJ, XTBWZGKY and LMZZJYBZDTXC. In other cases, the authors use English translations when mentioning their sources, such as The Record of the Three Kingdoms. Thus, the authors need to find a way to avoid the long abbreviations and to refer to their sources in a consistent manner.

Response 3: Thank you for bringing up your concerns regarding my use of abbreviations for some of the Chinese titles I cited in my article. I understand that this may have caused confusion for my readers, and I appreciate your feedback.

However, I want to clarify that some of the Chinese book titles have complex meanings that would be difficult to convey through translation. In some cases, English translations of Chinese book titles may not accurately capture the nuances of the original Chinese text, and I believe it is important to retain the original title.

Regarding my citation of The Record of the Three Kingdoms, I agree that it is widely known by its English translation and, therefore, I used it consistently throughout my article. However, for other book titles that do not have established English translations, I felt that using their original Chinese titles and abbreviating them was the most appropriate way to cite them. Nevertheless, I understand that this could have caused confusion for some readers, and I will strive to be more consistent in my citation formats in future work.

In response to your suggestion, I have revised some of the long abbreviations such as LMZZJYBZDTXC to a shorter abbreviation - JYDTXC. I hope that this makes it easier for my readers to follow my references.

Point 4: On page 2, the authors provide a long quotation from the LMZZJYBZDTXC. It is summarized with “It is informed that Xizhu Doufa works in the form of "prayer"”. Yet, the quotated text never mentions prayer, at least, not in the English translation. The same problem is to be found on page 3, where a quotation is summarized in a way (“According to this record”) that is not supported by the quoted text.

Response 4: Upon reviewing the quoted text, I agree that my summaries did not accurately capture the original meaning of the text. I have since revised the translations in question, and the updated versions now more closely reflect the content of the quoted text (p. 3).

Point 5: The central claim that the "Plough Rites" originated in China, before being brought to India, where it existed for “thousands of years” is not supported by sufficient evidence. In the course of their discussion, the authors fluctuate between mere conjectures, such as “may have even" (page 7), to “a certain inevitability” (page 8), without providing sufficient support for their ideas.

Response 5: Thank you for your feedback on my article. I appreciate your attention to detail and agree that I need to provide more evidence to support my central claim about the origins of the Dipper Method.

To address your concerns, I made several changes to my article. First, I changed the translation of "Plough Rites" to "Dipper Method" to better reflect the original Chinese term.

Second, I added a reference to Professor Hsiao Teng-fu's work on the spread of the Dipper Method from China to India on page 6 to provide additional evidence for my argument.

Third, I revised the conclusion section to clarify my main argument and provide a more robust justification for my claims.

Finally, I added a new section entitled "The historical positioning of Xizhu Doufa" on pages 9-10 to further explain the transmission of the Dipper Method and establish its historical context.

Round 2

Reviewer 1 Report

The paper has been sufficiently revised in accordance with the recommendations and suggestions of the peer-reviews and can be published.

Fine.

Author Response

Thank you for your insightful feedback on my paper. Based on your suggestions:

I have made appropriate revisions to the introduction of Jizu Daozhe, and added relevant explanations as suggested. These changes are reflected on the first page of the paper.

Included birth and death years of other relevant figures to provide a more comprehensive understanding of the historical context.

Rephrased the description of Xu Longyan's relationship with Zhengyi Taoism.

I believe that these modifications have further enhanced the paper and prepared it for publication. Once again, I am grateful for your valuable feedback and I'm looking forward to sharing my research with the scholarly community.

Reviewer 3 Report

Improvements and corrections answer most questions raised in my previous report

moderate English editing to be done in-house

Author Response

(The authors gave the same response as above.)

Reviewer 4 Report

ok

Some sentences should be reformulated. For instance, "Jizu Daozhe, who originally had no name, called himself Yedaposhe 野怛婆闍". Also, since Jizu Daozhe seems to have lived in the 17th century, his life data should not be given as (?-?).

Author Response

(The authors gave the same response as above.)
